



# Numerical Prediction of the Aerodynamics and Aeroacoustics

# of a Horizontal Axis Wind Turbine

Wen-Yu Wang[a],*, Yuh-Ming Ferng[b]

5       [a] Department of Greenergy, National University of Tainan, Tainan City, Taiwan

[b] Department of Engineering and System Science, National Tsing Hua University,
Hsinchu City, Taiwan

**Correspondence:** Wen-Yu Wang (wywang@mail.nutn.edu.tw)



## Abstract

This study used low-frequency-based numerical methods to predict noise radiating from rotating horizontal axis wind turbine (HAWT) blades. ANSYS FLUENT was used to calculate flow parameters in the vicinity of blade surfaces, as required for the Ffowcs Williams–Hawkings (FW–H) equation. The numerical model was validated against the experimental results from the National Renewable Energy Laboratory Phase VI wind turbine
blades. The coupling analysis was integrated with four Reynolds-averaged Navier–Stokes turbulence models and the FW–H equation under different boundary conditions. The SST k-ω and V2f turbulence models produced results in agreement with the available experimental pressure-coefficient and relative-velocity-distribution data. An INER 25-kW HAWT was employed to predict noise frequency distribution at nine points from the tower on the
windward and leeward sides under different operating conditions. Noise frequency distributions on the windward and leeward sides showed little differences, whereas those on the left and right sides with respect to the tower were different owing to wind-shear influence. The peak amplitude of the noise was inversely proportional to the increasing distance from the tower but proportional to the wind and rotation speeds.

**Keywords:** Aeroacoustic, Aerodynamic, CFD, Horizontal Axis Wind Turbine, Noise Frequency Distribution, Turbulence Model



## 1. Introduction


Global climate change and the increasing price of fossil fuels have forced many countries to develop renewable energy systems, which generally include solar, wind, geothermal, hydroelectric, and ocean energies. Wind energy provides a particularly exciting opportunity for generating renewable energy because of its abundance. Because of its

favorable geographical location and the influence of monsoons in Taiwan, there is abundant wind energy onshore and offshore. This is why Taiwan power companies are committed to developing wind power systems. Although small wind turbines are suitable for cities or suburbs to provide household electricity, the radiated noise can be heard by residents in adjacent neighborhoods and can cause serious environmental problems. These problems can

be divided into two categories: mechanical noise generated by frictional vibrations in the gearbox and aerodynamic noise generated by the rotating blades.

Mechanical noise can be eliminated by using sound-absorbing materials, but aerodynamic noise cannot be reduced by the same mechanism. Thus, aerodynamic noise remains the main source of wind turbine noise. It can be classified into three categories:

low-frequency noise, airfoil self-noise, and turbulent inflow noise. Low-frequency noise is generated when rotating blades interfere with the localized flow around the tower or when there are changes in wind speed or wakes, which are caused by other blades. The characteristic frequency range of low-frequency noise is ~10–200 Hz. A detailed description of airfoil self-noise and turbulent inflow noise can be found in previous studies (Brooks et al.,

1989; Hashem et al., 2017). Low-frequency noise radiating from a wind turbine is caused mainly by inflow noise; its long-distance propagation induces environmental noise issues.

Several researchers have investigated wind turbine noise prediction and reduction. For instance, Lighthill et al. (1952) derived the acoustic wave equation using fluid mechanics and mathematical models and predicted the far-field acoustics using experiments and

computational fluid dynamics (CFD). However, their theory is only applicable to turbulent flow generated by the surface of a stationary object. This method of simulating the flow field and the sound field separately is called acoustic analogy. Thereafter, Ffowcs Williams and Hawkings derived the FW–H equation based on the continuity equation and the Navier–Stokes equations to predict sound pressures. It was primarily derived using Lighthill's theory

for the accurate analysis of a sound field around moving objects in a flow field (Ffowcs et al., 1969). The noise radiated to the far field was predicted using the FW–H equation for two formulations: the original nonpermeable and permeable formulations.

Tadamasa and Zangeneh (2011) introduced wind turbine modeling for aerodynamic noise at low frequencies using the Reynolds-averaged Navier–Stokes (RANS) equations with

the SST k-ω turbulence model for aerodynamic calculation and the FW–H equation for aeroacoustics prediction. The researchers used the ANSYS/CFX CFD solver to predict the



noise of the National Renewable Energy Laboratory (NREL) Phase VI wind turbine blades. The numerical results using the FW–H equation for helicopter and aircraft rotors agreed very well with the experimental results obtained from an anechoic wind tunnel. The developed

FW–H equation was then used to determine the noise radiating from the NREL Phase VI wind turbine blades at various operating conditions (Sørensen et al., 2002). Filios et al. (2007) simulated blade noise using the FW–H equation, but only the surface thickness and loading noise sources were considered. Although the FW–H equation considers only monopole and dipole sound sources on the surface of an object, it can accurately predict the flow field.

Acoustic pressure predictions for the NREL downwind rotor have been presented for several cases, proving that broadband noise propagation dominates in low- and mid-frequency ranges. For instance, Arakawa et al. (2005) performed the first direct noise simulation of a full wind turbine blade using large eddy simulation (LES) to minimize tip blade noise. The sound pressure level decreased in the high-frequency domain for the

ogee-type tip shape. Mo and Lee (2011) numerically predicted the characteristics of aerodynamic noise generated by rotating wind turbine blades using incompressible LES. The far-field aerodynamic noise for frequencies below 500 Hz was modeled using the FW–H analogy. As the wind speeds increased, marked tonal noises were observed over the entire frequency region of 300–500 Hz. The simulated aerodynamic performance agreed well with

the experimental data.

Wasala et al. (2015) validated the aeroacoustic analysis of an airfoil using LES. An LES of the CART-2 wind turbine blade section in an annular domain was conducted to estimate the far-field noise due to unsteady aerodynamic loading. At high frequencies, the directivity had a typical dipole shape, but it was more omnidirectional at low frequencies. Maizi et al.

(2018) used ANSYS/CFX to simulate the noise radiating from the NREL Phase VI horizontal axis wind turbine (HAWT). The FW–H analogy was used to predict the aeroacoustic noise, which was then compared to experimental results. The aeroacoustic noise simulation used a transient flow field, unsteady RANS equations, and detached eddy simulation (DES). These methods were used to calculate the near-flow fields of different blade shapes. An acceptable

agreement between DES results and experiments was found. The controlled conditions of the experiment eliminated local atmospheric environmental effects, such that reliable and comprehensive full-scale wind turbine data were obtained, which are widely used to verify the accuracy of numerical results (Ghasemian et al., 2015; Zhang et al., 2016). In this study, the model was established using the airfoil blade data provided by a previous study (Giguere

et al., 1999). The simulation results and the experimental results of the NREL Phase VI wind turbine were used to simulate the flow field using different turbulence models (Sørensen et al. 2002; Giguere et al. 1999).

A commercial CFD solver, ANSYS FLUENT, was utilized to simulate and compute aerodynamic flow characteristics and parameters, which were required as input data for the



FW–H equations. The flow solver was validated on a model of the INER 25-kW wind turbine. This paper focused on the low-frequency noise of the turbulent boundary layers due to the passage of the blade. The surface pressure coefficients for two different inflow conditions with a freestream velocity of $U_\infty$ = 7.15 m/s were compared with the experimental data in a previous study (Nilay Sezer-Uzol, Ankur Gupta & Lyle N. Long, 2009). This study sought to

validate the numerical methodology for predicting noise radiating from the small wind turbine blades to the far field.

## 2. Method

### 2.1 Numerical method

The blade rotation of a HAWT produces complex aerodynamic characteristics. The geometry of the blades and the tip speed cause changes in the turbulent wake and tip vortices. These phenomena affect not only the thrust force required for rotation but also the flow field and sound field.

The CFD method involves discretizing the solution region into many finite volumes or mesh cells. There are three main numerical methods for discretization. The finite difference method discretizes the classical form of the partial differential equation (PDE). The finite element method discretizes the weak form of the PDE. The finite volume method (FVM) discretizes the conservation form of the PDE. The FVM has the relative advantage of being

mathematically straightforward and is particularly excellent for problems where quantity conservation is vital. Herein, the powerful CFD solver FLUENT was chosen, and FVM was employed to solve the Navier–Stokes equations. As 3D aerodynamic numerical simulations require substantial computing resources and time, the 3D model must be simplified. The basic assumptions in this paper are as follows:

1. The fluids are Newtonian and incompressible.

2. The effects of gravity and buoyancy are ignored.

3. There is a no-slip boundary condition on the blades.

4. The blades are assumed to have a smooth surface, and the effect of blade roughness on the flow field is ignored.

The governing equations are as follows:

Continuity equation:

$$\frac{\partial \rho}{\partial t} + \nabla \cdot (\rho v) = 0, \tag{1}$$





where

$\rho$ = density

$\nu$ = velocity

Momentum equation:


$$\frac{\partial}{\partial x_i}\left(\rho u_i u_j\right) = -\frac{\partial P}{\partial x_i} + \frac{\partial \tau_{ij}}{\partial x_i} + \rho g_i \qquad (2)$$

$$\tau_{ij} = (\mu + \mu_t)\frac{\partial u_i}{\partial x_j}, \qquad (3)$$

where

$P$ = static pressure

$\tau_{ij}$ = stress tensor

$\mu$ = viscosity

$\mu_t$ = turbulent viscosity

$\rho g_i$ = body force

$x_i$ = $x$-coordinate

$x_j$ = y-coordinate

$u_i$ = $x$-component velocity

$u_j$ = $y$-component velocity



## 2.2 Turbulence model equations

Turbulent flow is quite complicated. LES can more accurately simulate turbulent flow characteristics and wake vorticity than RANS simulations. However, the computing resources
required are great and the computing time is relatively long. Therefore, selecting appropriate



turbulence models, mesh sizes, and boundary conditions is an important consideration to effectively and accurately predict a turbulent flow field. The most commonly used turbulence model in CFD is the standard k-ε model, based on the eddy-viscosity concept. This study intends to compare four RANS-based turbulence models (standard k-ε, realizable k-ε, SST

k-ω, and V2f). All four models are based on the standard k-ε model. The models produce similar main flow structures but there were significant differences in the local flow fields. The equations are as follows (ANSYS Inc., 2018).

1. Standard k-ε:

The model solves for two variables: the turbulence kinetic energy and the rate of dissipation of turbulence kinetic energy. The model is popular for engineering applications owing to its favorable convergence rate and relatively low memory requirements.

Turbulence kinetic energy**:**


$$\frac{\partial}{\partial t}(\rho k) + \frac{\partial}{\partial x_i}(\rho k u_i) = \frac{\partial}{\partial x_j}\left[\left(\mu + \frac{\mu_t}{\sigma_k}\right)\frac{\partial k}{\partial x_j}\right] + G_k + G_b - \rho\epsilon - Y_M + S_k \qquad (4)$$

Dissipation rate**:**

$$\frac{\partial}{\partial t}(\rho\epsilon) + \frac{\partial}{\partial x_i}(\rho\epsilon u_i) = \frac{\partial}{\partial x_j}\left[\left(\mu + \frac{\mu_t}{\sigma_\epsilon}\right)\frac{\partial\epsilon}{\partial x_j}\right] + C_{1\epsilon}\frac{\epsilon}{k}(G_k + C_{3\epsilon}G_b) - C_{2\epsilon}\rho\frac{\epsilon^2}{k} + S_\epsilon \qquad (5)$$


2. Realizable k-ε:

The model is more accurate than the standard k-ε model in predicting the distribution of the dissipation rate. It also better predicts boundary layer characteristics in separated and recirculating flows.


Turbulence kinetic energy:

$$\frac{\partial}{\partial t}(\rho k) + \frac{\partial}{\partial x_j}(\rho k u_j) = \frac{\partial}{\partial x_j}\left[\left(\mu + \frac{\mu_t}{\sigma_k}\right)\frac{\partial k}{\partial x_j}\right] + G_k + G_b - \rho\epsilon - Y_M + S_k \qquad (6)$$

Dissipation rate:

$$\frac{\partial}{\partial t}(\rho\epsilon) + \frac{\partial}{\partial x_j}(\rho\epsilon u_j) = \frac{\partial}{\partial x_j}\left[\left(\mu + \frac{\mu_t}{\sigma_\epsilon}\right)\frac{\partial\epsilon}{\partial x_j}\right] + \rho C_1 S\epsilon - \rho C_2 \frac{\epsilon^2}{k+\sqrt{\nu\epsilon}} + C_{1\epsilon}\frac{\epsilon}{k}C_{3\epsilon}G_b + S_\epsilon \qquad (7)$$

3. SST k-ω:



The SST turbulent model incorporates original k-ω model of Wilcox in the inner region of the boundary layer, while transitioning to the Standard k-ε model in the outer region and free shear flows.The model works well for flow prediction both near and far from the wall and can be used for a wide range of Reynolds numbers. It is more nonlinear and more difficult to converge than the k-ε model. However, the model provides a better prediction of

flow separation than most RANS models, which accounts for its accuracy in adverse pressure gradients. On account of its good performance, SST k-ω model is frequently applied in the aerodynamics.

Turbulence kinetic energy:


$$\frac{\partial}{\partial t}(\rho k) + \frac{\partial}{\partial x_i}(\rho k u_i) = \frac{\partial}{\partial x_j}\left[\Gamma_k \frac{\partial k}{\partial x_j}\right] + \tilde{G}_k - Y_k + S_k \tag{8}$$

Specific dissipation rate:

$$\frac{\partial}{\partial t}(\rho \omega) + \frac{\partial}{\partial x_i}(\rho \omega u_i) = \frac{\partial}{\partial x_j}\left[\Gamma_\omega \frac{\partial \omega}{\partial x_j}\right] + G_\omega - Y_\omega + D_\omega + S_\omega \tag{9}$$

4. V2f:

     The V2f model describes the anisotropy of the turbulence intensity in the turbulent

boundary layer using two new equations for the velocity scale and the elliptic relaxation. The model is essentially an extension of the k-ε model, with the computational advantage of using the eddy-viscosity concept to close the transport equations. The model can be integrated to the wall, eliminating the need for damping or wall functions. The V2f model has been successful in accurately simulating a variety of non-equilibrium flows. For instance, it has

been applied to subsonic and transonic airflow around airfoils (Kalitzin, 1999), flows featuring adverse pressure gradients and bluff bodies (Durbin, 1995), as well as three-dimensional turbulent boundary layers driven by pressure around a wing-body junction (Parneix et al., 1998).

V2:

$$\frac{\partial}{\partial t}\left(\rho \overline{v^2}\right) + \frac{\partial}{\partial x_i}\left(\rho \overline{v^2} u_i\right) = \frac{\partial}{\partial x_j}\left[\left(\mu + \frac{\mu_t}{\sigma_k}\right)\frac{\partial \overline{v^2}}{\partial x_j}\right] + \rho k f - 6\rho \overline{v^2}\frac{\epsilon}{k} + S_{\overline{v^2}} \tag{10}$$

     f:






$$f - L^2 \frac{\partial^2 f}{\partial x_j^2} = \frac{(C_1-1)}{T}\left(\frac{2}{3} - \frac{\overline{v^2}}{k}\right) + C_2 \frac{2\mu_t S^2}{\rho k} + \frac{5\overline{v^2}}{Tk} S_f,$$ (11)

where $G_k$ represents the generation of turbulence kinetic energy due to the mean velocity gradients. $G_b$ is the generation of turbulence kinetic energy due to buoyancy. $G_\omega$ represents

the generation of $\omega$. $Y_M$ represents the contribution of the fluctuating dilatation in compressible turbulence to the overall dissipation rate. $C_1$, $C_2$, $C_\mu$, $C_{1\varepsilon}$, $C_{2\varepsilon}$, and $C_{3\varepsilon}$ are constants. $\sigma_k$ and $\sigma_\varepsilon$ are the turbulent Prandtl numbers for k and $\varepsilon$, respectively. $S_k$, $S_f$, and $S_\varepsilon$ are user-defined source terms. $\Gamma_k$ and $\Gamma_\omega$ represent the effective diffusivity of k and $\omega$, respectively. $T = k/\varepsilon$, and $L$ is a length scale.


### 2.3 Aeroacoustic formulation

The FW–H method, the most general form of the Lighthill acoustic analogy, was used to
predict the far-field noise. The Lighthill acoustic analogy is a combination of the continuity equation and momentum conservation equation and is appropriate for the prediction of sound generated by rigid bodies in arbitrary motion. The FW–H equation is a rearrangement of the continuity equation and the Navier–Stokes equations into an inhomogeneous wave equation with sources of sound. The FW–H equation is derived by obtaining the equation that can be
applied within the entire unbounded domain, both inside and outside the control surface using generalized functions to describe the flow field. The FW–H equation can be written as follows.

$$\frac{1}{a_0^2}\frac{\partial^2 p'}{\partial t^2} - \nabla^2 p' = \frac{\partial^2}{\partial x_i \partial x_j}\left[T_{ij} H(f)\right] - \frac{\partial}{\partial x_i}\left\{\left[P_{ij} n_j + \rho u_i (u_n - v_n)\right]\delta(f)\right\}$$

$$+ \frac{\partial}{\partial t}\left\{\left[\rho_0 v_n + \rho(u_n - v_n)\right]\delta(f)\right\}$$ (12)

where $u_n$ is the fluid velocity in the direction normal to the integration surface, $v_n$ is the normal velocity of i, $\delta(f)$ is the Dirac delta function, $H(f)$ is the Heaviside function, and $\rho_0$ and $a_0$ are the density and the speed of sound in an unbounded space, respectively.

A formal solution of the FW–H equation is obtained using the free space Green's
function, retaining only thickness and loading source terms. The acoustic solution discussed in this paper is given by the following equation:





$$4\pi {a_0}^2 [\rho(x,t) - \rho_0] =$$

$$\frac{\partial^2}{\partial x_i \partial x_j} \int \left[\frac{T_{ij}J}{r|1-M_r|}\right] dV_C - \frac{\partial}{\partial x_i} \int \left[\frac{P_{ij}n_j}{r|1-M_r|}\right] dS + \frac{\partial}{\partial t} \int \left[\frac{\rho_0 U_n}{r|1-M_r|}\right] dS \qquad (13)$$


where $M_r$ is the Mach number vector, which is the local surface velocity vector divided by the freestream sound speed. The right side of the equation contains three sound sources; their denominators are all related to the Mach number. In the case of a high Mach number, the three sound sources cannot be ignored. The first source is the Lighthill quadrupole term for

the integral of the control volume, related to the turbulent flow generated by the fluid passing through the blades. The second term is the dipole or loading noise, which is related to the force exerted on the fluid by the surface of the blades. The third term is the monopole or thickness noise, related to the flow velocity perpendicular to the surface of the blades.

**2.4 CFD model**

FLUENT was used to calculate the aerodynamic flow parameters required as inputs to the FW–H equations. The flow solver was validated on the NREL Phase VI HAWT wind turbine blade, as shown in Fig. 1. This wind turbine blade model is widely used in validating numerical codes for predicting aerodynamic performance owing to the availability of

experimental data for various operating conditions (Tadamasa et al. 2011; Sørensen et al. 2002). The NREL Phase VI wind turbine is two-bladed and has a 10.7-m diameter with a power rating of 25 kW. It is a stall-regulated wind turbine with full-span pitch control. The blade has an S809 airfoil cross-section designed especially for use in a wind turbine, as shown in Fig. 1. The top is assumed to be a cylinder, the shaft position is at 30% of the chord

length, the wingspan is 5.029 m, and the top is a plane. The nacelle and tower behind the wind turbine are ignored in the overall computational domain (Fig. 2) to simplify the model, and are assumed to not affect the results (Hand et al. 2001). The blade surface and its surroundings require a finer mesh to accurately measure flow separation (especially close to the surface) because of stagnation and separation problems. The far field can be calculated

using a coarser mesh. The far-field domain is defined as a cylinder with a diameter of 12 m, and the height of the blade is 12.192 m. The rotor is a short cylinder with a radius of 6 m and a thickness of 1.5 m, and it formed the boundary area for blade rotation.

Before flow field analysis, a mesh dependency study must be conducted to ensure that the influence of numerical errors caused by mesh inaccuracy is minimized. Thus, the surface

pressure coefficient ($C_p$) on the blade was compared with available experimental data and LES results (Nilay Sezer-Uzol, Ankur Gupta & Lyle N. Long, 2009) to validate the simulation and ensure sufficient mesh density.



Two different meshes were used, one coarse and one fine with 2,105,622 and 5,814,235 cells, respectively. A structural mesh was used in the near-wall region to effectively control

the quality and quantity of the mesh on the blade surface. Fig. 3 depicts the surface pressure-coefficient distribution on three spanwise sections at 30%, 47%, and 80% of the blade span with a 7 m/s inflow velocity. The simulated $C_p$ agreed well with the experimental data and LES results.

The standard k-ε turbulence model was used in the mesh dependency study. The $C_p$

calculated using the coarse mesh was lower in the leeward position of the front end than the experimental results. If the $C_p$ was significantly increased, the calculated velocity was relatively small. Compared with the experimental values and LES data, the simulated pressure was relatively large. When the simulation used the fine mesh, the pressure coefficient agreed well with the LES model and the experimental data. Only the front end

saw predicted pressure coefficients that were too high. There was no obvious difference between the results of the standard k-ε model and the LES model when the fine mesh was employed. In the latter parts of this study, differences were seen for other turbulence models. Therefore, this study used the fine mesh to obtain more accurate simulation results.


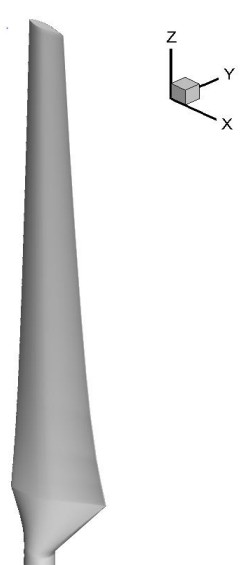

**Fig. 1.** NREL Phase VI blade.



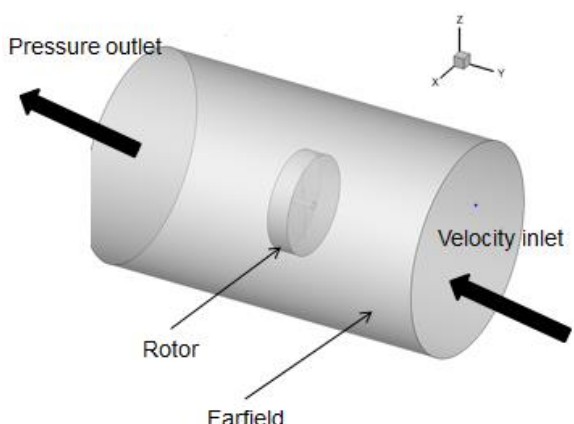


**Fig. 2.** Computational domain.

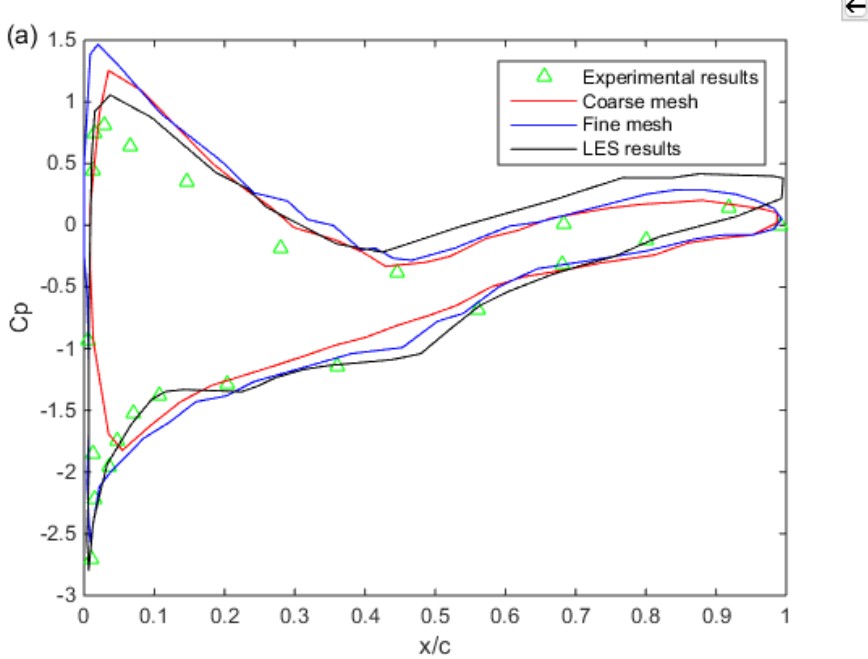



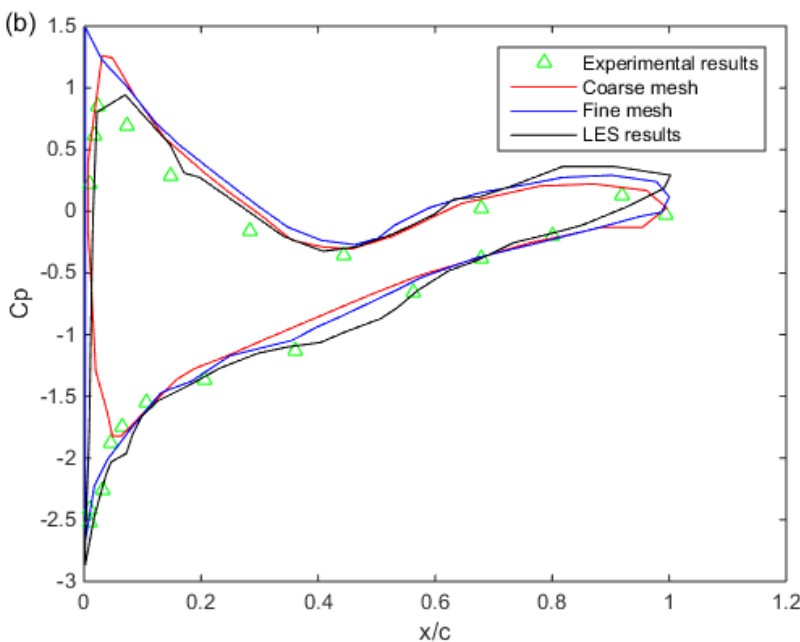


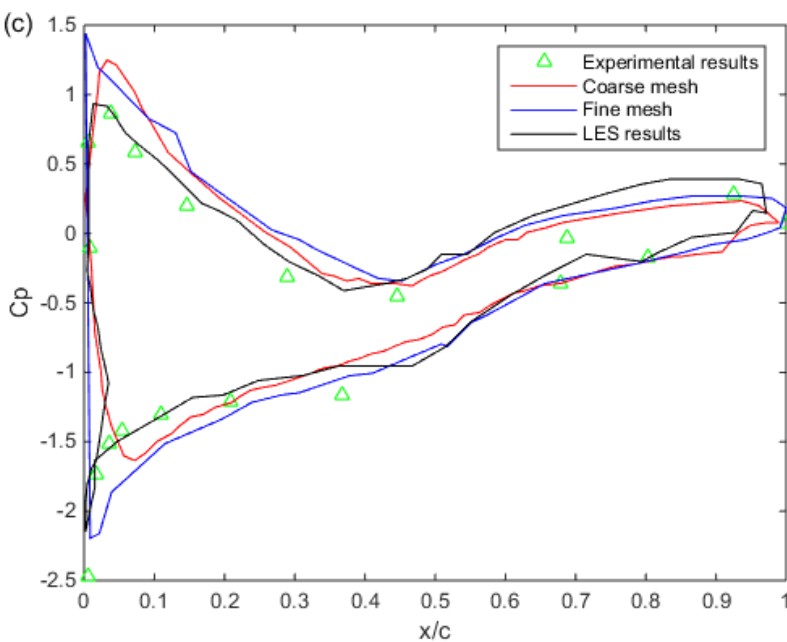



**Fig. 3.** Comparison of the experimental and numerical pressure-coefficient distributions at a wind speed of 7 m/s at (a) 30%, (b) 47%, and (c) 80% of the chord length for different relative heights.

**2.5 Problem description**

The INER 25-kW wind turbine is a three-bladed rotor. The geometry of the blade is based on the S809 airfoil. The diameter of the blades is $D = 10.7$ m, and the tower height is $H = 25.3$ m (Fig. 4). The model setup of the mesh dependency study and validation was followed, and the domain was divided into the far field and the rotation area. The blades

reside in the rotation area, the origin is the center point, and the wind speed is set in the positive $y$-axis direction.

The diameter of the rotor is 14 m, and the far-field radius is 25.3 m. The tower and the nacelle are ignored to reduce the computation time. Because the INER 25-kW is a three-blade HAWT, the mesh requires a large number of cells, particularly close to the blade surfaces and

in between the blades. Therefore, a structural mesh was used on the blade surface to control the thickness of the boundary layer in order to accurately predict stagnation and separation.

The mesh utilized in the dependency study was improved for flow field and sound field prediction, including the structural mesh near the blade surface. The far field of the computational domain only required a coarse unstructured mesh. The total number of cells

was 10,284,331. Due to the fine mesh, selecting the correct turbulence model to save computing resources while still achieving the required accuracy was a very important process.

In terms of sound field prediction, positions 0, 25, 50, and 70 m from the wind turbine bottom were utilized. An additional monitoring point is at 31 m, which is based on the

international standard monitoring distance (rotation diameter/2 + tower height). The sound and noise simulations were conducted for three different operating conditions, as shown in Table 1.



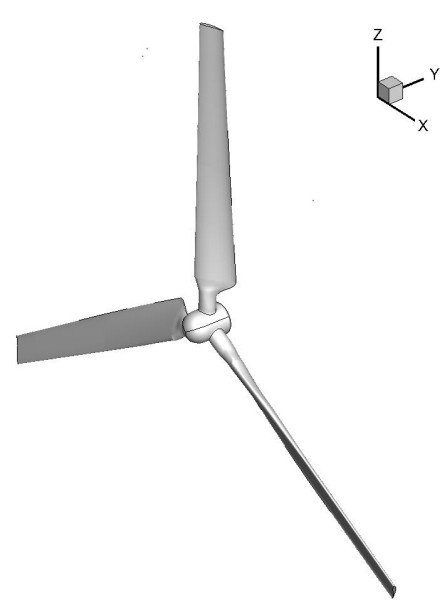

**Fig. 4.** INER 25-kW wind turbine blades.

**Table 1.** Wind noise simulation under different operating conditions

| Case | Velocity (m/s) | Pitch (˚) | Rotation speed (m/s) |
|---|---|---|---|
| Case 1 | 6 | 6 | 44.05 |
| Case 2 | 12 | 18 | 59.13 |
| Case 3 | 18 | 33 | 60.85 |


## 3. Results and discussion

**3.1 Aerodynamics results**

Because of advances in computational technology, the interest in computational
aeroacoustics has grown. The use of CFD in wind turbine design has been limited to date



because of inaccuracies of traditional RANS turbulence models when predicting highly unsteady features that are crucial in noise prediction.

Four turbulence models were studied. Simulations were performed for two inflow conditions (U∞ = 7 and 15 m/s), and the results were compared with the LES results and experimental data from a previous study. Fig. 5 shows the relative velocity distribution at a wind speed of 7 m/s at 30% chord distance for the different turbulence models. The standard k-ε model is suitable for the simulation of high Reynolds–number flows. However, the flow

velocity in the near-wall region was impacted by viscous effects, and the flow had a low Reynolds number. Therefore, the surface effects must be solved using a wall function, but this model did not consider the rotational effect of the fluid when predicting the separated flow field and vortices. Thus, the flow field near the surface must be solved using the wall function in the realizable k-ε model. The standard k-ε model increased the rotational effect of the fluid

with its empirical equation by changing the turbulent viscosity. The separated flow field calculated by the realizable k-ε model is more accurate than that of the standard k-ε model. Although the V2f model is based on the standard k-ε model, two additional equations are added to address the effects of Reynolds stresses. The low Reynolds–number flow in the near-wall region is considered in the V2f model, so there is no need for additional wall

functions.

    The SST k-ω turbulence model is a commonly used two-equation eddy-viscosity model. It is a hybrid model combining the Wilcox k-ω and the k-ε models. A blending function, F1, activates the Wilcox model near the wall and the k-ε model in the free stream. This ensures that the appropriate model is utilized throughout the flow field. Fig. 5 shows only a slight

difference between the standard k-ε model and the other turbulence models, which may be because of the low wind speed. In Fig. 6, the relative velocity distribution predicted by the standard k-ε model only exhibits a slight difference from the other models. Compared with the four turbulence models in this study, the LES model more clearly grasps the vortex dynamics in the dissipation of the wake. Overall, however, the simulation results in this study

were close to the LES model results from the previous study. Fig. 6 shows the relative velocity distribution at a wind speed of 7 m/s and 80% chord distance. Similar simulation results were observed at different chord distances in Figs. 5–7. The fluid flow at the blunt blade tip was significantly lower when using the standard k-ε model, as compared to the other turbulence models because the $y^+$ value was in the range of 30–90. If more accurate

simulation results are required, the mesh size near the wall must have a $y^+$ of less than one. Thus, the mesh size must be finer, and a low Reynolds–number turbulence model or enhanced wall treatment method must be employed. If a wall function is used, the fluid in the near-wall region is not as accurately predicted, which may increase errors.

    Fig. 8 compares the $C_p$ distributions of different turbulence models at a wind speed of

7 m/s with the experimental data. The $C_p$ of the standard k-ε model was significantly lower,



indicating that the static pressure was too high and the speed was too low. In the simulation results of the realizable k-ε model, the $C_p$ at the blade tip was too high, the speed results were too high, and the pressure was too low. The results of the SST k-ω model and the V2F model were closest to the experimental data.

Fig. 9 shows the relative velocity distributions at a wind speed of 15 m/s and 30% chord distance for the different turbulence models. Although the standard k-ε model was not completely consistent with the LES model or experimental results, the trend was similar. In the realizable k-ε model, the velocity was too high in the blunt area on the front of the blade, which was inconsistent with the experimental results. This is because the turbulent viscosity

was a coefficient in the standard k-ε model. The physical quantity of rotation was added with the realizable k-ε model and was calculated using the square root of the fluid strain rate, which is the average vorticity of the fluid. In the case of a single condition (i.e., a single static flow field or a single swirling flow field), the realizable k-ε model was better than the standard k-ε model in predicting separation or fluid rotation. However, the simulation domain

contained static and rotating regions at the same time, so there were nonphysical phenomena when using the realizable k-ε model. This occurred where the differences to the experimental $C_p$ values were more pronounced.

Fig. 10 shows the relative velocity distribution at a wind speed of 15 m/s and 47% chord distance for the different turbulence models. The simulation results of the realizable k-ε

model were unreasonable. Although the standard k-ε model cannot effectively simulate the vortex characteristics of the wake area behind the blade, the overall trend was still reasonable. The SST k-ω and V2f models successfully simulated the two wakes behind the blades.

Fig. 11 shows the relative velocity distribution at a wind speed of 15 m/s and 80% chord distance for the different turbulence models. The simulation trends are similar in Figs. 9 and

10. The standard k-ε model can simulate the entire region, but the realizable k-ε model is not suitable for wind turbines. The V2f and SST k-ω models have more accurate simulation results as verified via pressure-coefficient experimental data.

Fig. 12 shows the simulated and experimental pressure-coefficient distribution at a wind speed of 15 m/s and different chord distances. The pressure-coefficient distribution of the

realizable k-ε model has a much larger error than that of the other turbulence models. Thus, the results of the standard k-ε model in the windward area of the front end are relatively inaccurate, whereas those of the SST k-ω and V2f models in the same area are accurate. The resuts of three turbulence models in the leeward area has high pressure, but there is no significant difference from the experimental results. Thus, the SST k-ω and V2f models are

superior for simulating the flow field.



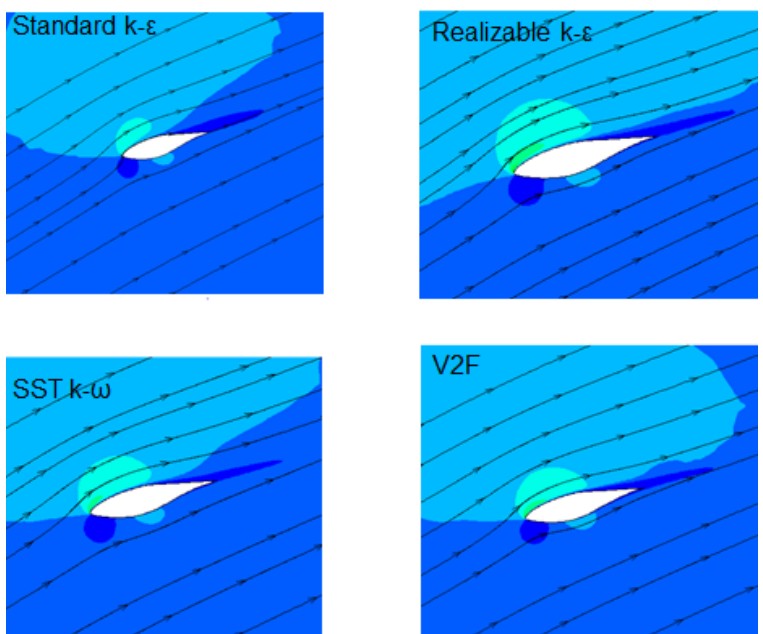

**Fig. 5.** Relative velocity distribution at a wind speed of 7 m/s and 30% chord distance for different turbulence models.


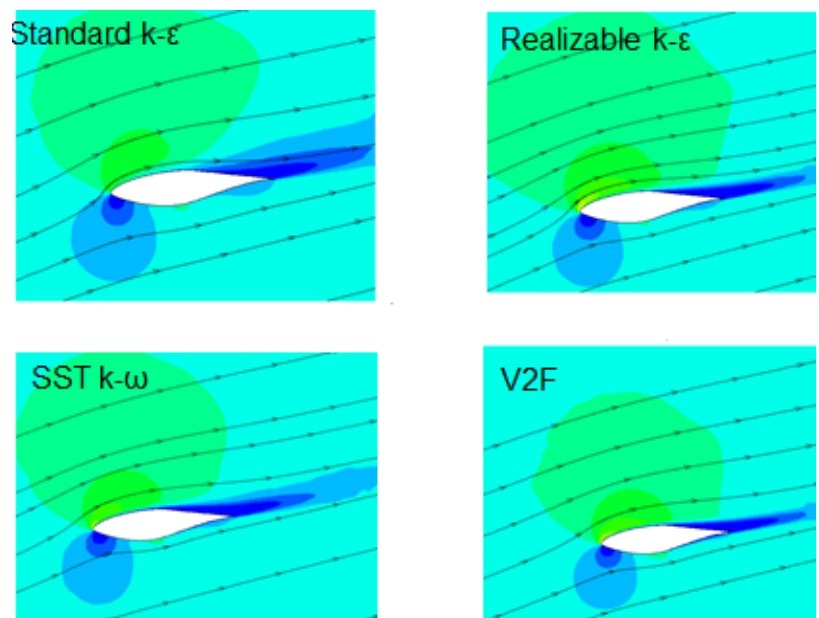

**Fig. 6.** Relative velocity distribution at a wind speed of 7 m/s and 47% chord distance for different turbulence models.





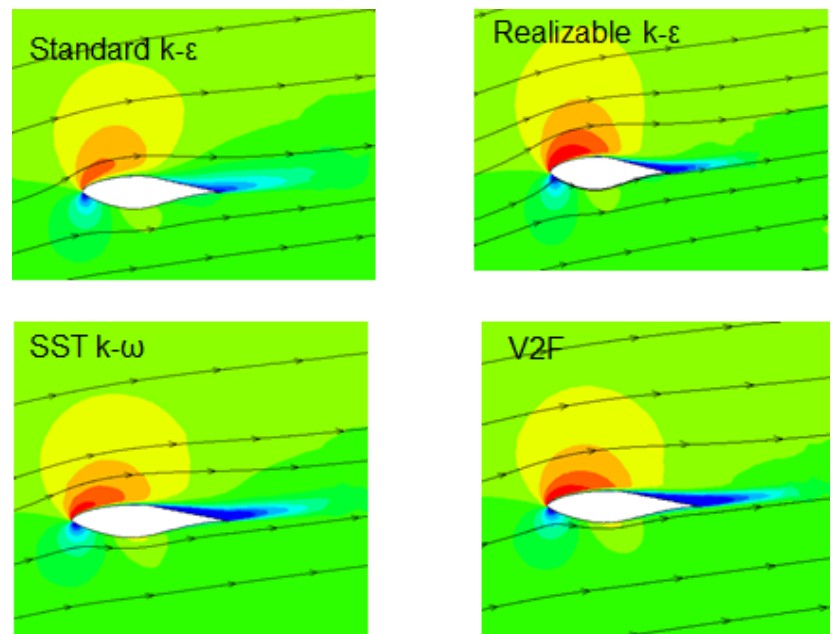


**Fig. 7.** Relative velocity distribution at a wind speed of 7 m/s and 80% chord distance for different turbulence models.



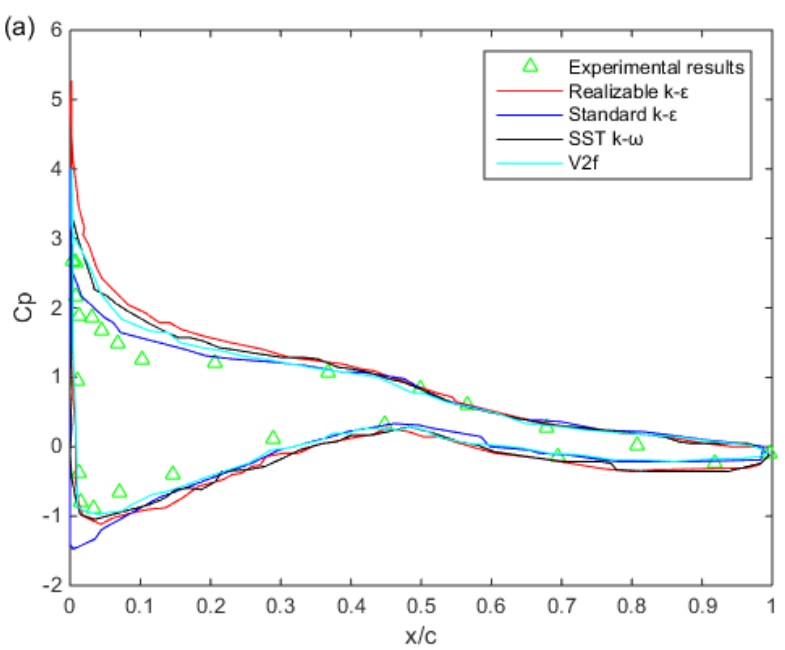

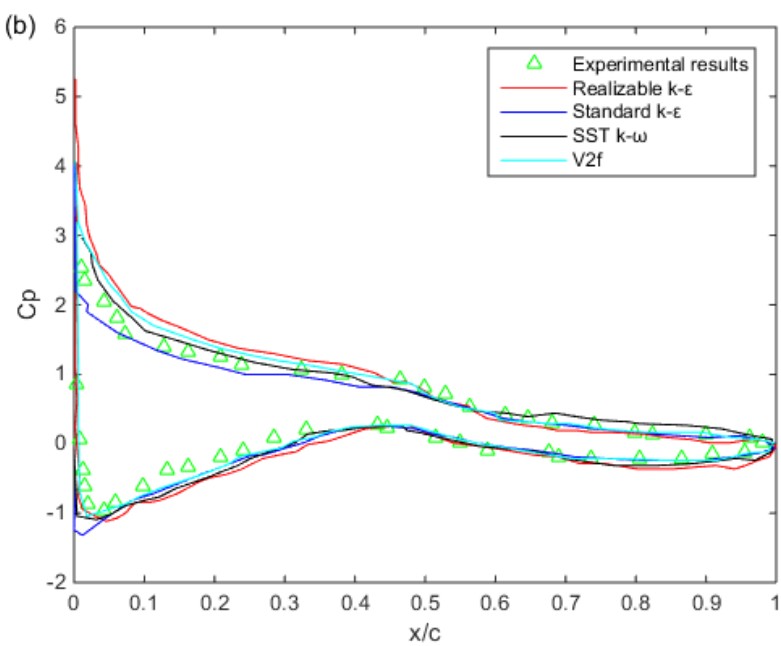

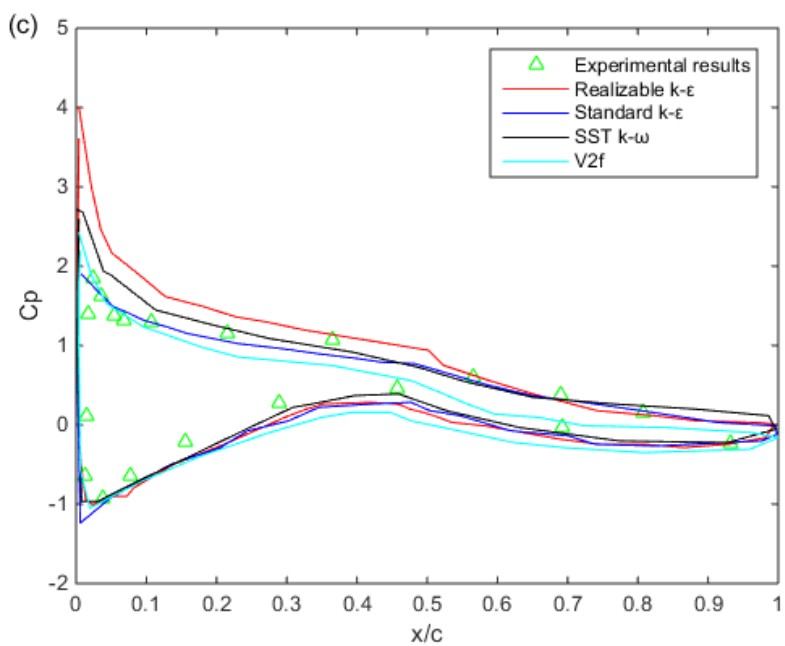

**Fig. 8.** Pressure-coefficient distributions at a wind speed of 7 m/s and (a) 30%, (b) 47%, and (c) 80% chord distances for experimental data and different turbulence models.

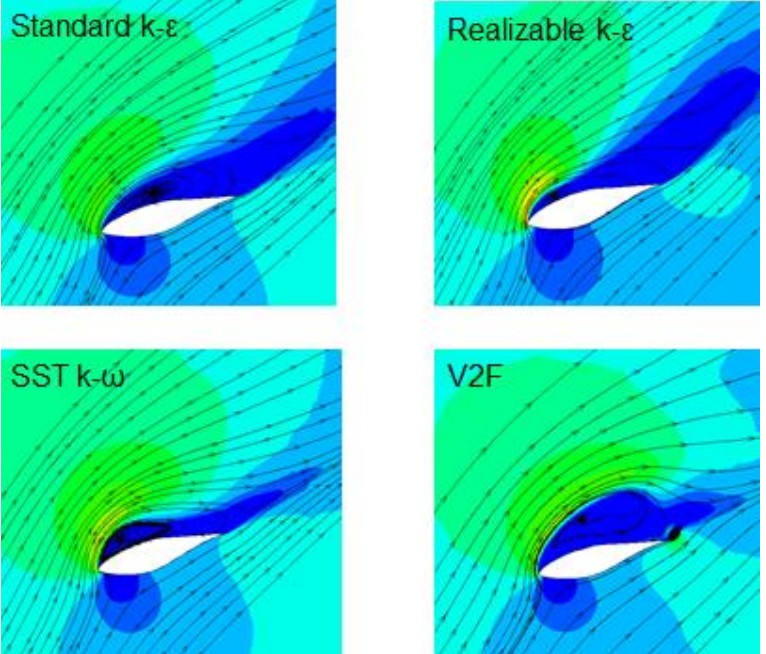






**Fig. 9.** Relative velocity distribution at a wind speed of 15 m/s and 30% chord distance for different turbulence models.

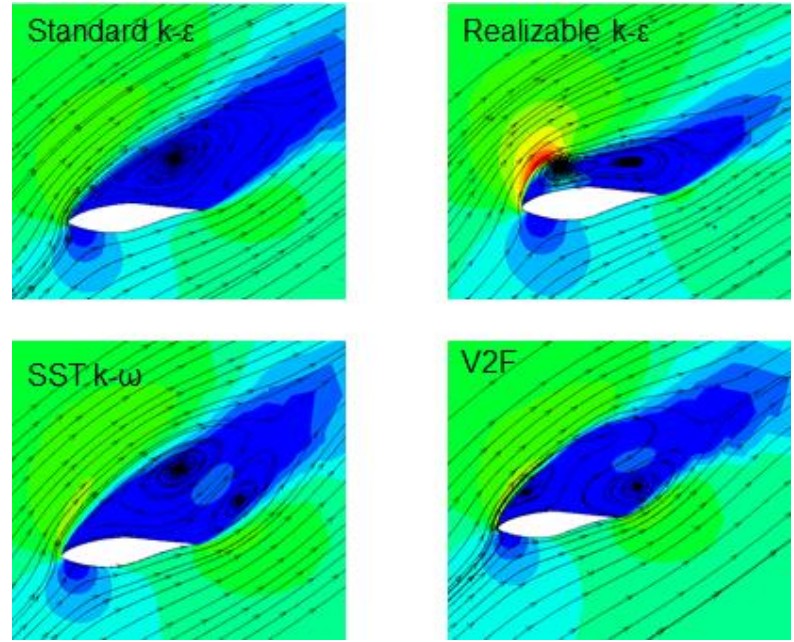

**Fig. 10.** Relative velocity distribution at a wind speed of 15 m/s and 47% chord distance for
different turbulence models.

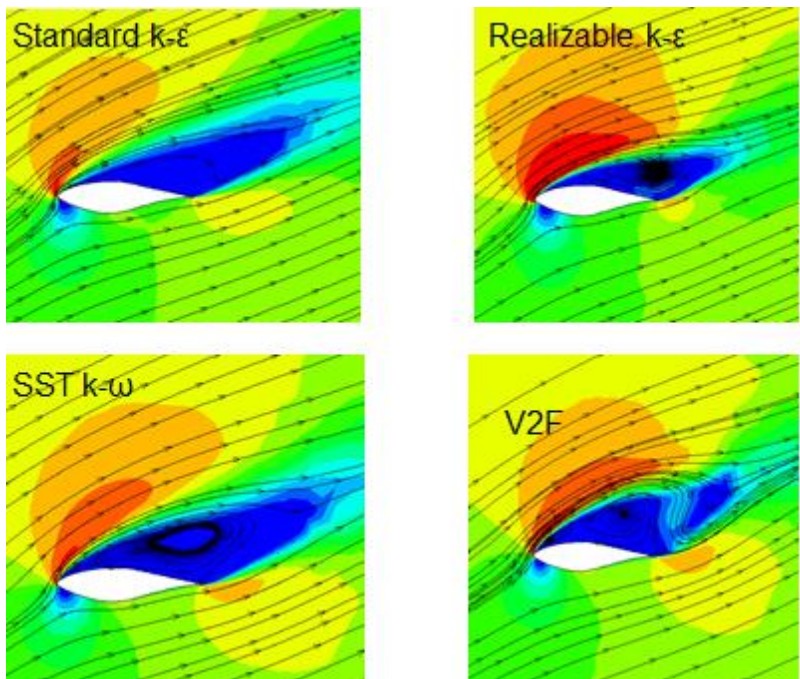

**Fig. 11.** Relative velocity distribution at a wind speed of 15 m/s and 80% chord distance for different turbulence models.

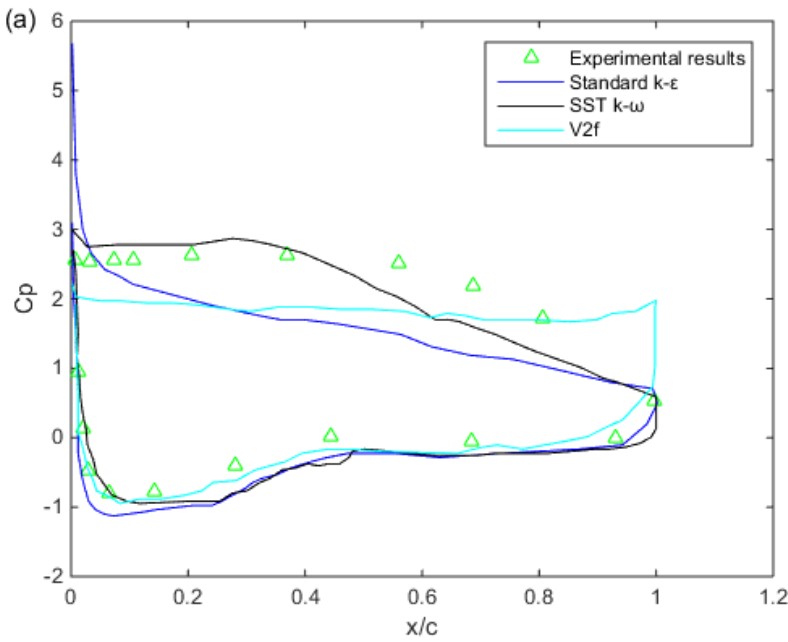






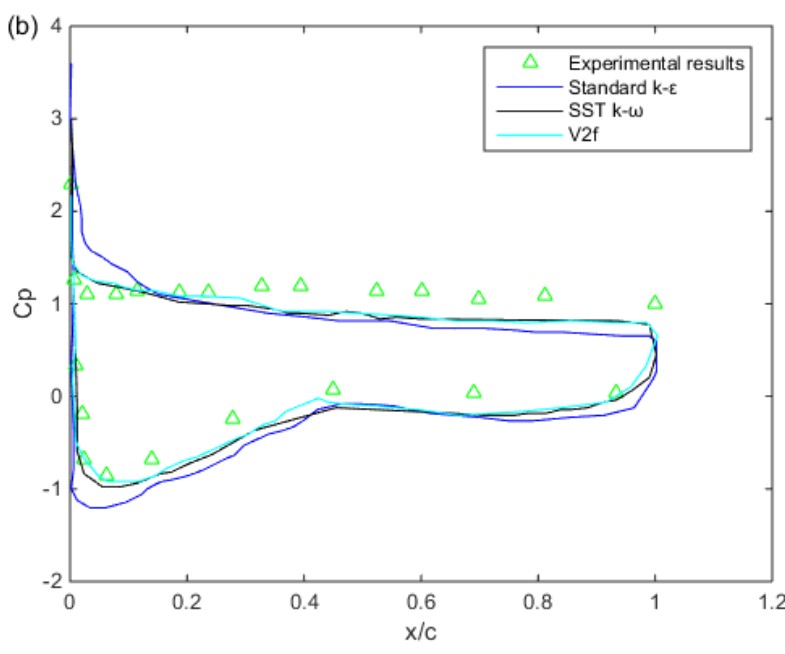

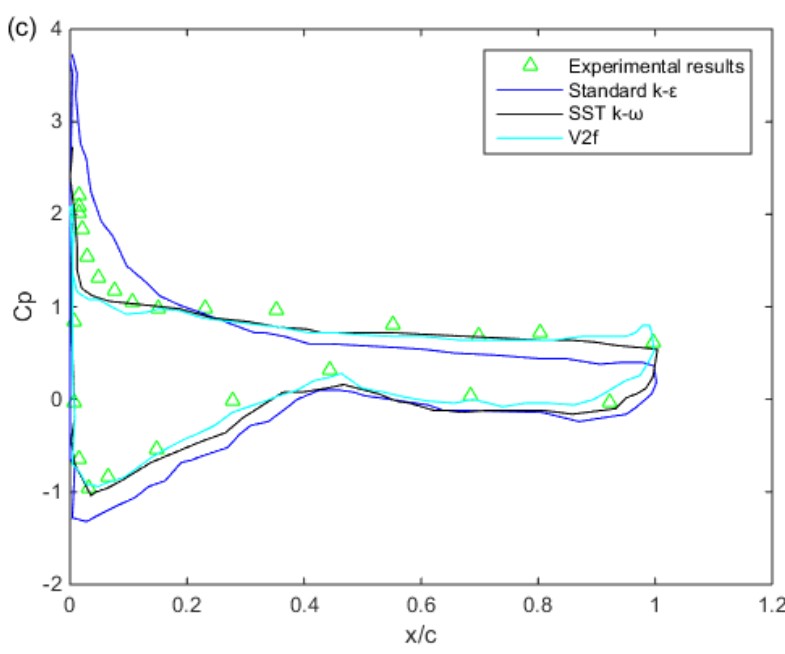





**Fig. 12.** Pressure-coefficient distributions at a wind speed of 15 m/s and (a) 30%, (b) 47%, and (c) 80% chord distances for experimental data and different turbulence models.

### 3.2 Validation of aeroacoustic results

The frequency range that can be heard by the human ear is 20–20,000 Hz, and the dominant low-frequency aerodynamic noise of a turbine blade resides in the same range. Therefore, the standard k-ε, SST k-ω, and V2f models were used to simulate the sound and noise field. The boundary conditions were a wind speed of 7 m/s, a rotation speed of 72 rpm, a time step of 0.000416, and a frequency domain of 1200 Hz. Fig. 13 compares the three turbulence models and the previous LES results (Tadamasa and Zangeneh, 2011). The aerodynamic noise trends of the four different turbulence models all decreased exponentially, with a frequency range of 1–1200 Hz. The low frequencies fall between 10 and 50 dB, whereas those of 100–200 Hz have the most severe impact. Compared with the results of the LES model, the standard k-ε turbulence model combined with the FW–H equation underestimated the noise by ~5 dB. The main advantage of the FW–H equation is its ability to separate each source term and determine which type of noise is dominant. Consequently, the SST k-ω and V2f turbulence models were more consistent with the aerodynamic LES data.

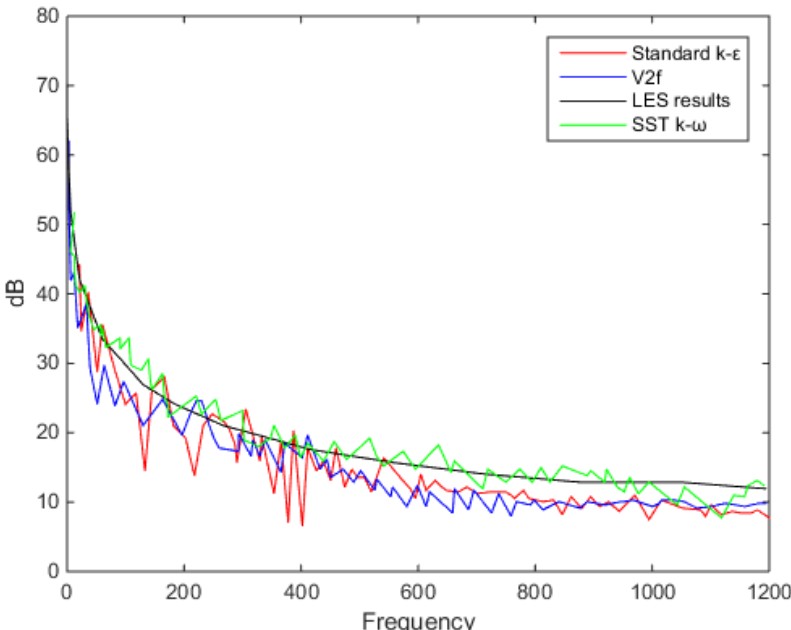

**Fig. 13.** Comparison of the sound fields when using different turbulence models.

520

### 3.3 Noise simulation of the INER 25-kW wind turbine

This study established a complete set of wind turbine simulation processes for both the flow field and the sound field. From the flow field analysis, the relative velocity and $C_p$ distribution comparisons to experimental data, and the verification of the noise simulation, 525  the SST k-ω turbulence model was selected. In this study, nine points 0, 25, 31, 50, and 70 m away from the tower were taken as noise monitoring points under three cases: (1) wind speed 6 m/s, pitch = 6°, (2) wind speed 12 m/s, pitch = 18°, and (3) wind speed 18 m/s, pitch = 33°. Fig. 14 shows the dB–frequency distribution under a wind speed of 12 m/s and a pitch of 18°. Monitoring points 1 and 2 were the noise frequency distribution 31 m to the leeward and to 530  the windward side, respectively. Under these operating conditions, the decibels of the low-frequency noise at 0–200 Hz were ~25–40 dB, and the noise frequency distributions on the windward and leeward sides were similar, possibly due to not being greatly affected by the blade speed. The difference in sound and noise frequency distributions was caused by the wind shear on the left and right sides. In the three cases, the noise frequency distributions of 535  the windward and leeward sides at the same distance all showed this phenomenon, though there was little difference between the windward and leeward sides. Therefore, the results of





the leeward side were selected for discussion.

Fig. 15 shows the sound and noise simulation of the monitoring point under the tower at three different wind speeds. At higher operating output power (Case 1: 5.266 W; Case 2: 27.0159 W; and Case 3: 29.5496 W), the influence of aerodynamic noise generated by wind shear is higher. The average noise at low frequencies (0–200 Hz) was ~40 dB. At the lowest operating output power, the average decibel was ~25 dB. Fig. 16 shows the noise simulation for three different wind speeds on the leeward side at a distance of 25 m from the wind tower. The declining trend of the decibel number in Case 1 was more obvious than in the other two cases. The sound pressure waves from lower sound sources may be the cause, as their decay mechanism is much faster than that of pressure waves from higher sound sources and becomes more pronounced with distance. In the low-frequency region, Case 1 was nearly 0 dB, whereas Case 2 attenuated to 25 dB and Case 3 was 35 dB.

Figs. 17 and 18 show the receiving points 50 and 70 m from the tower, respectively. The decibel attenuation was the same as that in Fig. 16, although the attenuation mechanism increased with distance and when under high-noise operating conditions. In the case of low sound source operation, the decline was noticeable. With increasing distance, the decibel number of the monitoring point ~25 m away dropped to 0 dB. For the noise prediction in Case 2, the decibel number at 50 m was ~25 dB and was ~15 dB at 70 m. In Case 3, the decibel number at 50 m was ~30 dB and was ~20 dB at 70 m. The noise simulation results were also consistent with the flow field situation. In the case of high flow velocity and negative angle of attack, the vortex generated by the wake behind the blades was the main source of aerodynamic noise, and the expected decibel number was thus higher.

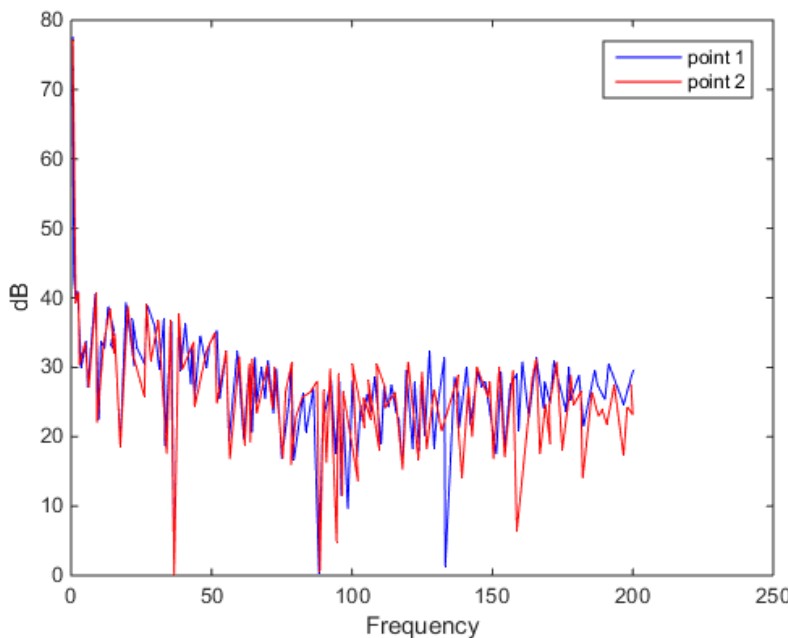


**Fig. 14.** dB–frequency distributions under a wind speed of 12 m/s and a pitch of 18° (Case 2).

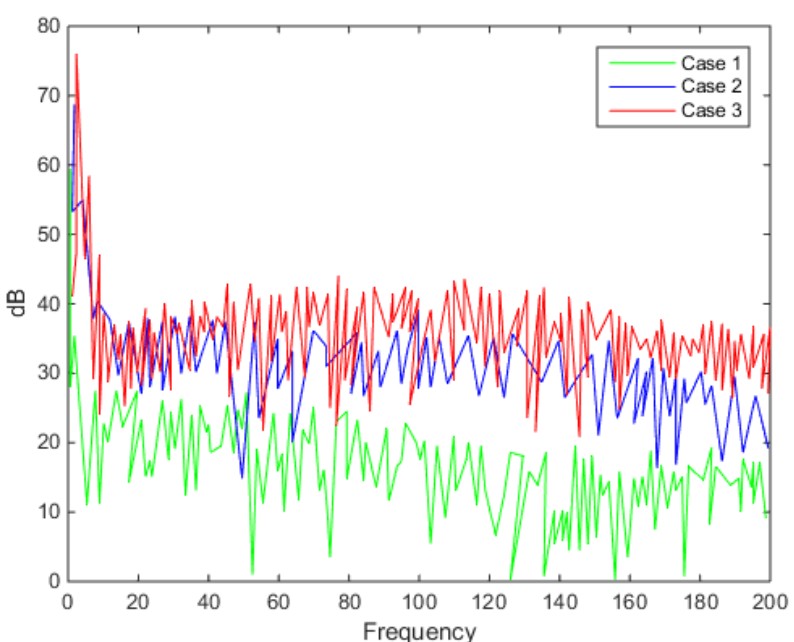

**Fig. 15.** Noise frequency distribution at 0 m away from the tower.

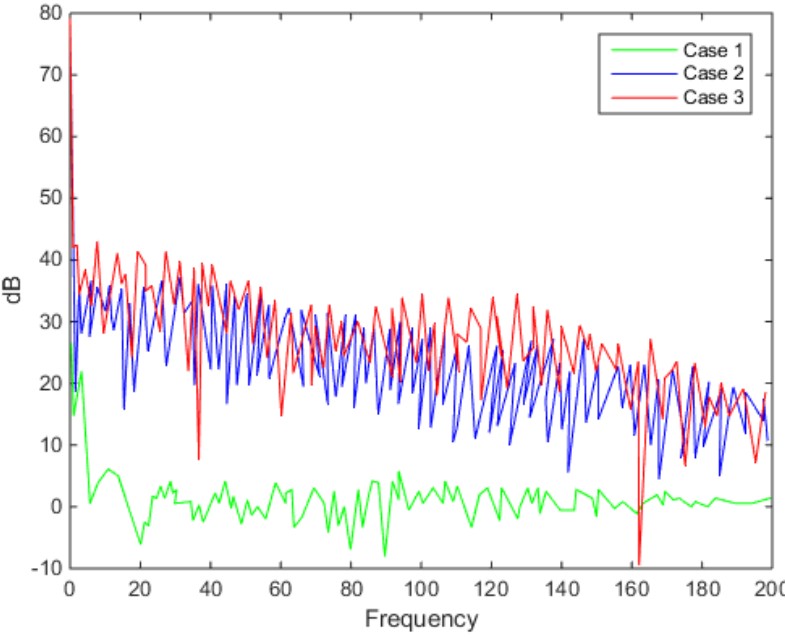

**Fig. 16.** Noise frequency distribution at 25 m away from the tower.

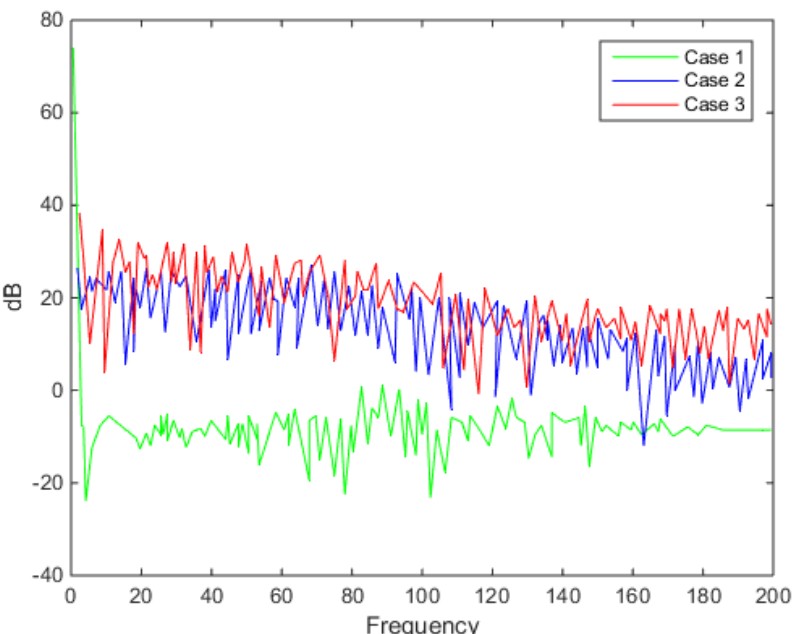

**Fig. 17.** Noise frequency distribution at 50 m away from the tower.

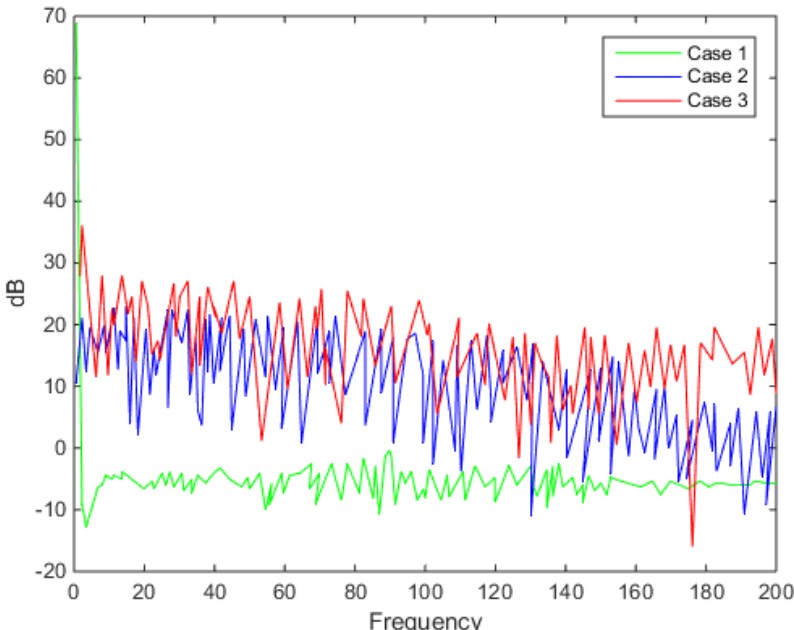

**Fig. 18.** Noise frequency distribution at 70 m away from the tower.


## 4. Conclusions

ANSYS FLUENT was used to simulate aerodynamic flow characteristics for different turbulence models. The aerodynamic simulation results were validated using experimental measurements of the NREL Phase VI wind turbine. The results obtained for RANS

turbulence models were similar to those for LES. The slight difference was due to the relative velocity distribution in the wake region. When verifying the simulated pressure coefficients, the main difference from the experimental data was seen for the blade front part that exhibited flow separation, whereas the pressure-coefficient distribution predicted for the blade rear part was very similar to the experimental data. The pressure coefficients calculated

using the realizable k-ε model had the largest difference from the experimental results, especially near the blade front part. In Case 2, with high flow velocity, the difference in the wake from the experimental data was more pronounced than in the low-velocity flow in Case 1. Within the relative velocity analysis, the difference in the distribution of wake regions was more significant. Although the phenomena predicted by the turbulence models varied, there

was no significant difference in the overall trend. From the analysis of the pressure coefficients and the secondary results in the sound field analysis, most turbulence models



showed results similar to the experimental data, except for the standard k-ε model in the blade front part. Therefore, in the simulation of wind turbines, although the flow field and wake distribution in LES can provide better accuracy, it comes at an extremely high

computing cost. The V2f and SST k-ω turbulence models were also investigated. The predicted flow separation, wake vortex, overall flow field, and sound field characteristics were similar to the results of the LES and the experimental data. The results also showed that the peak amplitude in decibels is inversely proportional to the increasing distance from the tower but is proportional to the wind and rotation speeds. As a results, low computational cost

turbulence models can significantly reduce the calculation time, so that a wind turbine noise simulation process was developed in this study. The geometric shape of the wind turbine blades have a significant impact on the aerodynamic and aeroacoustic. The goal of reducing noise can be achieved by changing the blade design and installing noise reduction devices. Future studies will focus on extending these calculations to evaluate noise reduction devices

and a wider range of the sound field around the small wind turbines.

## Competing interests

The contact author has declared that none of the authors has any competing interests.


## Acknowledgments

Support from the Atomic Energy Council (AEC), Taiwan is acknowledged.

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
