# Peer review of "Numerical Prediction of the Aerodynamics and Aeroacoustics"

_Wind Energy Science, 2023_

## Author Comment (AC1)

**Reply to the Reviewer #1**

The paper presents a numerical method based on a commercial code to predict aerodynamic characteristics and noise emissions of HAWT. The CFD solution is coupled with FWH method to assess the noise spectrum at the observer locations. Different turbulence models are tested, and their results are compared with LES and experimental acquisitions. The paper faces a very interesting problem related to the annoyance of wind turbines located near populated areas. The main aspects of the noise prediction method are touched by the authors, but the description of the single step is quite shallow and not complete.

Authors Reply: The authors would like to thank the reviewer for the time to review our paper. The comments that the reviewer provided have contributed to the enhancement of our paper. We have taken the opportunity to make several improvements in the text, in order to strengthen the paper. A list of point-by-point replies to the reviewer's comments is reported in the following.

Comment 1: For instance, the numerical method description is a general discussion of basic CFD concepts without a deeper discussion of the motivation behind setup choices. Classical PDE equations are reported with some typos (e.g. in Eq. 2 the time derivative term is missing), and also the turbulence model description is too detailed: references to the different model formulation should be enough. Same story for the FWH formulation.

Authors Reply: Thanks for pointing out. Deeper discussion of the motivation behind setup choices have been added in the Numerical method section. Also, the PDE equations were revised in the Method section and we did not add any more references based on reviewer's suggestion.

Comment 2: Concerning the numerical model validation by using NREL HAWT, some important aspects of numerical simulation are missing: numerical schemes for diffusive and convective fluxes, detailed description od BCs., discussion about convergence criteria and so on. Moreover, is not crestal clear by looking at Fig. 3 that fine mesh performs better in terms of accuracy. Could the authors better explain their conclusions?

Authors Reply: Numerical schemes for diffusive and convective fluxes, detailed description on BCs., discussion about convergence criteria has been added in the Numerical method section. The conclusions of the comparison with the experimental

and numerical  $C_p$  distributions have been added. The  $C_p$  calculated using the coarse mesh was lower in the leeward position of the front end than the experimental results. The calculated velocity was relatively small when  $C_p$  was considerably increased. Compared with the experimental values and LES data, the simulated pressure was relatively large. When the simulation used the fine mesh,  $C_p$  agreed well with the LES model and the experimental data. Furthermore, the  $C_p$  of the fine mesh near x/c = 0agreed with the LES data. The predicted pressure coefficients were observed to be considerably high in the front end and trailing end. When the fine mesh was employed, no obvious difference was observed in the front end and trailing edge between the results of the standard k- $\varepsilon$  model and LES model. We added the detailed conclusion in the CFD model section.

Comment 3: Moving to the INER 25-kW turbine, some aspects of the operating conditions are not so clear: why the rotational speed is expressed in m/s? Should it read rad/s or rpn? When the authors discussed the aerodynamic results, they compared the different turbulence model, without discussing the results in detail. Could the author make a thorough discussion of this? Finally, the comparisons on Fig.12 show some discrepancies, could the authors comment on that?

Authors Reply: Thanks for pointing out. We have modified the rotational speed in rpm. The discussion of the comparison with the different turbulence model and the discrepancies in Fig. 13 in the new manuscript were added in the Aerodynamics results section. In these four model, the velocity was too high in the blunt area on the front of the blade, which was inconsistent with the experimental results predicted by realizable k-ɛ model. This is because the turbulent viscosity was a coefficient in the standard k-ɛ model. The physical quantity of rotation was added with the realizable k- $\varepsilon$  model and was calculated using the square root of the fluid strain rate, which is the average vorticity of the fluid. Standard k-& model is the general used in CFD to simulate mean flow characteristics for turbulent flow conditions. However, it does not calculate the flow field with high accuracy, especially in displaying reverse pressure gradients and strong curvature in flow field or blades. The SST- k-ω model provides a better prediction of flow separation and recirculation than most RANS based models, which accounts for its accuracy in adverse pressure gradients. The SST  $k-\omega$ turbulence model showed itself suitable for the numerical simulation of small scale wind turbines (Rocha et al., 2016; Akar et al., 2019). The V2f model has been successful in simulating a variety of non-equilibrium flows. In conclusion, the SST k- $\omega$  and V2f models are superior for simulating the flow field of small scale HAWT.

Comment 4: Concerning the noise prediction section, t is not clear how the CFD simulations used for noise predictions are performed. Do they rely on steady or unsteady simulations? Also, the FWH setup is not completely described: where the FWH surface is placed? Which is the sampling rate of the FFT?

Authors Reply: The flow field and the sound field from blades depend on unsteady simulation. We added the information in Aerodynamics results section. The FW-H surface was placed on the blades. According to the Nyquist-Shannon sampling theorem, the sampling rate must be at least twice the maximum frequency present in the signal. In this case, the maximum frequency in the frequency domain is 1200 Hz. Therefore, the minimum required sampling rate would be 2400 Hz. Therefore, the sampling rate of the FFT is approximately 1 / 0.000416 = 2403.85 Hz. We added the discussion in Verification of aeroacoustic results section. Thanks for pointing out.

Comment 5: Moreover, is quite strange to see noise spectra with negative value in dB (that is under the human hearing threshold). In addition, is there a blade passing frequency in the spectra? If so, please discuss a bit on this aspect.

Authors Reply: With increasing distance, the decibel number of the monitoring point ~25 m away from the tower dropped to 0 dB in case 1, so we deleted all values below 0 dB and the lines that the noise often below 0 db in Noise simulation of the INER 25-kW wind turbine section.

Yes, in the field of engineering and rotating machinery, such as wind turbines, compressors, and fans, a phenomenon known as Blade Passing Frequency (BPF) can be observed in the spectra. We added some discussion in Noise simulation of the INER 25-kW wind turbine section.

Comment 6: Finally, an English revision of the wording is highly suggested.

Authors Reply: We regret there were problems with the English. The new manuscript has been carefully revised by a professional language editing to improve the grammar and readability. Thank you for the valuable comment.

---

## Author Comment (AC2)

**Reply to the Reviewer #2**

The reviewed manuscript presents a study aimed at developing an affordable computational fluid dynamics (CFD) method using Reynolds-averaged Navier-Stokes (RANS) simulations to accurately predict aerodynamic noise from wind turbine rotors. Addressing rotor noise is a topic of great significance, as reducing rotorgenerated noise can minimize turbine curtailment and increase Annual Energy Production (AEP). However, the manuscript requires several improvements to enhance clarity, address inconsistencies, and strengthen the analysis and discussion.

Authors Reply: The authors would like to thank the reviewer for the time to review our paper. The comments that the reviewer provided have contributed to the enhancement of our paper. We have taken the opportunity to make substantial improvements in the text, in order to strengthen the paper. A list of point-by-point replies to the reviewer's comments is reported in the following.

General Comments:

1. The manuscript lacks clarity in several sections, making it difficult to discern the findings and identify the computational model that produces the best results. The language and spelling need improvement throughout the manuscript.

Authors Reply: The new manuscript did substantial improvements to enhance the clarity in several sections. Moreover, the new manuscript has been carefully revised by a professional language editing to improve the grammar and readability. Thank you for the valuable comment.

2. The discussion on the computational domain and convergence criteria of the simulations is insufficient. Additionally, important mesh characteristics and simulation performance details are missing, hindering a comprehensive evaluation of the simulation quality.

Authors Reply: The computational domain, convergence criteria of the simulations, and simulation performance were added in the Numerical method section. The coarse and fine mesh characteristics were added in Fig. 3 in the new manuscript based on reviewer's suggestion.

3. There is a mixing up of mesh diameter and radius, as well as the definition of rotor rotational speed in m/s.

Authors Reply: Thanks for pointing out. We have changed the rotational speed in rpm and the words required as above.

Specific Comments:

Validation of Numerical Setup (NREL-Phase VI):

 The authors conduct a mesh sensitivity study using two mesh sizes (~2m & ~6m) and compare the results with LES simulation and experimental data. However, there is no clear convergence observed within the results that are shown. The authors should provide a more detailed analysis and discuss the limitations of the mesh sensitivity study or conduct additional numerical experiments until a convergence can be observed.

Authors Reply: The  $C_p$  calculated using the coarse mesh was lower in the leeward position of the front end than the experimental results. The calculated velocity was relatively small when  $C_p$  was considerably increased. Compared with the experimental values and LES data, the simulated pressure was relatively large. When the simulation used the fine mesh, the simulated  $C_p$  agreed well with the LES model and the experimental data. Furthermore, the  $C_p$  of the fine mesh near x/c = 0 agreed with the LES data. We added some discussion and the limitations of the mesh sensitivity study in CFD model section. We already have two different meshes to conduct a mesh sensitivity study. We understand that the current mesh sensitivity study has not met your requirements. However, the subsequent validation process has shown minimal differences in the turbulence models results. We hope for your understanding. In the future, we will conduct more mesh sensitivity study based on this paper, using vertical-axis or horizontal-axis wind turbines.

2. The agreement of the LES simulations with experimental data towards the trailing edge is relatively poor. This discrepancy needs to be addressed and discussed in order to provide a comprehensive assessment of the simulation results.

Authors Reply: We have added some discussion of the discrepancy between LES and experimental data in CFD model section.

3. The discussion on the computational domain should be expanded to assess whether it adequately captures a fully developed rotor wake. Additionally, the convergence criteria for the simulations should be clearly described. Authors Reply: We observed that the velocity gradient at the outer boundary was observed to be zero, indicating that the domain adequately captures a fully developed rotor wake. Therefore, the domain is sufficient in our analysis. The discussion was included in the CFD model section. Also, the convergence criteria has been added in the Numerical method section.

Simulation of the INER 25kW Rotor:

1. The use of a 10m cell mesh raises concerns regarding the transferability of the previous mesh convergence study. The authors should address this issue and explain the rationale behind the selected mesh size.

Authors Reply: An unstructured inviscid mesh with 3.6 million tetrahedral cells clustered around the blades and tip vortices was created in the reference (Sezer-Uzol, N et al., 2009). In the reference, a total mesh of 9.6 million tetrahedral cells clustered around the blades and tip vortices was created for LES model in the whole domain and similar to the simulations in this paper (10m). The extra meshes are located outside the computational domain and do not affect the simulation results of the flow field near the wind turbine based on the reference. Detail 10m cells can be seen in the figure below. We added some discussion in Problem description section.

2. Figures 5 to 11 compare velocity distributions generated by four different turbulence models. However, the discussion of these results is qualitative, and it is difficult to draw meaningful conclusions. The authors should provide a more detailed analysis, discussing the strengths and weaknesses of each turbulence model in specific cases. Providing additional rotor performance characteristics such as rotor power, thrust or wake velocity distributions could help the reader to better assess the quality of the presented simulations.

Authors Reply: The meaningful detailed conclusions of four models were added in the Aerodynamics results section in the new manuscript. Standard k-E model is the most common used in CFD to simulate mean flow characteristics for turbulent flow conditions. However, it does not calculate the flow field with high accuracy, especially in displaying reverse pressure gradients and strong curvature in flow field or blades. The standard k- $\varepsilon$  model works better for simpler geometry (Farhadi et al., 2017). The SST-  $k-\omega$  model provides a better prediction of flow separation and recirculation than most RANS based models, which accounts for its accuracy in adverse pressure gradients. The SST k-w turbulence model showed itself suitable for the numerical simulation of small scale wind turbines (Rocha et al., 2016; Akar et al., 2019). The V2f model has been successful in simulating a variety of non-equilibrium flows. The realizable k-ɛ model is not suited for accurately predicting near-wall flows. It tends to overpredict turbulence levels near solid walls, especially in regions of adverse pressure gradients in Figs. 6-8 and Figs. 10-12. It is designed for isotropic and homogeneous turbulence and may not handle anisotropic or non-homogeneous flows well. Moreover, SST  $k-\omega$  and V2f model were agreed well with the experimental results at different chord distances.

3. Legends are missing in the velocity plots, which hampers understanding. The authors should include a clear legend to improve clarity.

Authors Reply: Thanks for pointing out. We have included the normalized legends in figures. 6-8 and figures. 10-12 in the new manuscript.

Noise Prediction and Discussion:

1. The authors should clarify that the results are "verified" against an LES simulation, rather than "validated," as the term "verification" is more appropriate.

Authors Reply: We have modified the words required as above.

2. The discussion on noise predictions could be strengthened by providing more detailed analysis and comparisons with experimental data or established noise prediction models.

Authors Reply: Based on reviewer's comment, we provided a new comparison with experimental data to strengthen the paper. We simulated the noise distribution (Fig. 20) below the tower and compare simulations with the experimental data in the

reference (Cheng et al., 2014) using A-weighting method (dB(A)). The location right below the tower is the closest to the sound source and the experimental result is less influenced by the surrounding. Hence, the predicted noise data in this paper agrees well with the experimental data. We added some discussion and Fig in the Noise simulation of the INER 25-kW wind turbine section.

**Conclusion:**

In conclusion, the manuscript requires substantial improvements to enhance clarity, address inconsistencies, and strengthen the analysis and discussion sections. The authors should focus on the methodology, providing important details regarding the computational domain, convergence criteria, mesh characteristics, and simulation performance. By addressing these issues and highlighting relevant findings, the manuscript can become eligible for publication.

Authors Reply: Thank you for your thorough review and time of this paper. We appreciate your suggestions for improvement. Based on your comments, we agree that there are several sections in the manuscript that require substantial enhancements to enhance clarity and strengthen the analysis. Additionally, we carefully review the paper to identify any inconsistencies in the data and discussion. We understand the importance of presenting accurate and reliable results, and we revised any discrepancies in the initial submission.

Thank you again for your time and valuable comments.